

# Modelling $SO_2$ conversion into sulphates in the mid-troposphere with a 3D chemistry-transport model: the case of Mount Etna's eruption on April 12, 2012.

Mathieu Lachatre[1,5], Sylvain Mailler[1,2], Laurent Menut[1], Arineh Cholakian[1], Pasquale Sellitto[3], Guillaume Siour[3], Henda Guermazi[3], Giuseppe Salerno[4], and Salvatore Giammanco[4]

[1]LMD/IPSL, École Polytechnique, Institut Polytechnique de Paris, ENS, PSL Université, Sorbonne Université, CNRS, Palaiseau, France.
[2]École des Ponts, Université Paris-Est, 77455 Champs-sur-Marne, France.
[3]Univ Paris Est Creteil and Université de Paris, CNRS, LISA, F-94010 Créteil, France
[4]Istituto Nazionale di Geofisica e Vulcanologia, Osservatorio Etneo, Catania, Italy.
[5]Presently at ARIA is now SUEZ, 8-10 rue de la Ferme 92100 Boulogne-Billancourt France

**Correspondence:** Mathieu Lachatre (mathieu.lachatre@suez.com)

**Abstract.** Volcanic activity is an important source of atmospheric sulphur dioxide ($SO_2$), which, after conversion into sulphuric acid, induces impacts on, among others, rain acidity, human health, meteorology and the radiative balance of the atmosphere. This work focuses on the conversion of $SO_2$ into sulphates ($SO_{4(p)}^{2-}$, $S(+VI)$) in the mid-tropospheric volcanic plume emitted by the explosive eruption of Mount Etna (Italy) on Apr. 12, 2012, using the CHIMERE chemistry-transport model. Since

volcanic plume location and composition depend on several often poorly constrained parameters, using a chemistry-transport model allows us to study the sensitivity of $SO_2$ oxidation to multiple aspects such as volcanic water emissions, transition metal emissions, plume diffusion and plume altitude. Our results show that in the mid-troposphere, two pathways contribute to sulphate production, the oxidation of $SO_2$ by OH in the gaseous phase (70 %), and the aqueous oxidation by $O_2$ catalyzed by $Mn^{2+}$ and $Fe^{3+}$ ions (25 %). The oxidation in aqueous phase is the faster process, but in the mid-troposphere, liquid water is

scarce, therefore the relative share of gaseous oxidation can be important. After one day in the mid-troposphere, about 0.5 % of the volcanic $SO_2$ was converted to sulphates through the gaseous process. Because of the nonlinear dependency of the kinetics in the aqueous phase to the amount of volcanic water emitted and on the availability of transition metals in the aqueous phase, several experiments have been designed to determine the prominence of different parameters. Our simulations show that during the short time that liquid water remains in the plume, around 0.4 % of sulphates manage to quickly enter the liquid

phase. Sensitivity tests regarding the advection scheme have shown that this scheme must be chosen wisely, as dispersion will impact both oxidation pathways explained above.

# 1 Introduction

Sulphate aerosols resulting from the conversion of volcanic sulphur dioxide ($SO_2$) have substantial effects on air quality, meteorology, rain acidity and the radiative balance of Earth atmosphere at local-to-global spatial scales, depending on the





specific volcanic activity (e.g. Langmann (2014); Sellitto et al. (2017); Pattantyus et al. (2018); Sellitto et al. (2191)). $SO_2$ is emitted from both anthropogenic and natural sources, volcanic emissions being the major contributors to the natural emissions. Volcanic emissions can be classified as coming from two broad classes of volcanic activity: passive degassing and explosive events. Passive degassing occurs permanently at many volcanoes. For example, Etna emits an estimated $530 \, \mathrm{kt \, yr}^{-1}$ of $SO_2$

annually, making it the eighth contributor to $SO_2$ emissions from passive degassing; the strongest contributor worldwide being Mount Kilauea (in Hawai, USA), which emits an estimated $2740 \, \mathrm{kt \, yr}^{-1}$ annually (Itahashi et al. (2021)). Explosive eruptions emit massive quantities of $SO_2$ into the atmosphere in a short period of time. Unlike passive degassing, which generates emissions close to the surface, explosive eruptions may emit $SO_2$ high above the volcanic vent, with the possibility of getting up to the stratosphere for massive eruptions such as El Chichón in 1982 (Pollack et al., 1983) or Mount Pinatubo (Philippines)

in 1992, or even the more moderate recent activity of volcanoes such as Raikoke (e.g. Kloss et al. (2021)).

While contributing to the air quality on a local-to-regional scale, the sulphate aerosols produced as a result of explosive volcanic activities represent an important natural radiative forcing as well and are therefore significant for climate studies. Pattantyus et al. (2018) give an extensive review of the oxidation processes of $SO_2$ in the Marine boundary layer for the case of Mount Kilaueaa, and list two main oxidation paths for this species, oxidation by hydroxyl radical (OH) in gas phase, and

oxidation in liquid phase (including oxidation by $H_2O_2$, $O_3$ and catalytic oxidation via $O_2$). However, its fate in volcanic plumes in the free troposphere still remains poorly understood, in part due to the difficulty of measuring these events. Multiple efforts have been carried out to understand and model sulphates formation within volcanic plumes, mostly at first phases of eruption events (Hoshyaripour et al., 2014, 2015; Roberts et al., 2019). Heard et al. (2012) have modelled the plumes from Kasatochi in 2008 , Mt. Sarychev in 2009, and Eyjafjallajökull in 2010 with the NAME dispersion model, with encouraging

results in reproducing the observed plumes of $SO_2$ and sulphates. Specific modelling work has been carried out using a 0D model (Galeazzo et al., 2018), and brought interesting insights into main oxidation pathways of $SO_2$: these authors highlight on the potential importance of the catalytic oxidation of $SO_2$ by $O_2$ with transition metals as catalysts. Pianezze et al. (2019) have explored the role of volcanic aerosols as Cloud Condensation Nuclei (CCN) and the evolution of their size distribution. In the free troposphere, volcanic particle size distribution is evolving to a coarser distribution as time goes by, and particles can

serve as CCN far from the vent (Sahyoun et al., 2019; Pianezze et al., 2019).

Regarding mid-tropospheric eruptions, the issue of aqueous chemistry, with the potential contribution of volcanic water emissions to the formation of an aqueous phase needs to be considered since there is the possibility that these emissions have an impact on sulphate formation for this portion of the atmosphere. In the case of boundary layer eruptions and passive degassing, the quantity of water vapour emitted by the volcano is typically much smaller than the background water vapour at

that level, while for stratospheric eruptions temperatures are too cold to allow the presence of liquid water. In particular, the question of sensitivity of sulphate formation to the volcanic emissions of water vapour is unanswered as of yet. In addition, the 0D study of Galeazzo et al. (2018) argues that aqueous oxidation of $SO_2$ catalyzed by transition metals may be a substantial, or even dominant, oxidation pathway, and that explosive eruptions themselves emit water vapour (possibly contributing to the formation of an atmospheric liquid phase) and transition metals. Another effect, not taken into account in the present study,

is the potential depletion of OH radicals due to its consumption by atmospheric halogen. Jourdain et al. (2016) conducted





a modelling study on the volcanic plumes of the Ambrym volcano (Vanuatu). They conclude that when taking into account halogen emissions, the lifetime of $SO_2$ relative to oxidation by OH increases by 36% compared to the same simulation without halogen emissions; the authors attribute this change to OH depletion.

Various pathways can lead to $SO_2$ [S(+IV)] oxidation to $SO_4^{2-}$ [S(+VI)]. In the gaseous phase, $SO_2$ can react with the OH
photochemically produced from ozone and water vapour:

$$O_{3(g)} \quad + \quad h\nu \quad \longrightarrow \quad O_2 + O(^1D) \tag{R1}$$
$$O(^1D) \quad + \quad H_2O \quad \longrightarrow \quad 2OH \tag{R2}$$

Gas-phase conversion of $SO_2$ by OH follows reactions R3-R5 (Seinfeld and Pandis, 2006):

$$SO_{2(g)} + OH + M \quad \longrightarrow \quad HOSO_{2(g)} + M \tag{R3}$$
$$HOSO_{2(g)} \quad + \quad O_2 \quad \longrightarrow \quad HO_2 + SO_{3(g)} \tag{R4}$$
$$SO_{3(g)} \quad + \quad H_2O \quad \longrightarrow \quad H_2SO_{4(g)} + M. \tag{R5}$$

Reaction R3, the limiting step in this mechanism, is relatively slow (the decay rate of $SO_2$ through this mechanism is estimated at $2.9 \pm 2.1\,\%\,h^{-1}$ during daytime for the remote marine conditions around Mt. Kilauea), therefore in presence of an aqueous phase liquid-phase conversion tends to dominate gas-phase conversion.

Since $SO_2$ is a soluble gas, aqueous-phase oxidation is also a possibility; the balance between liquid-phase and gas-phase concentrations being governed by the Henry's law:

$$[SO_2]_{aq} = H_{SO_2} p_{SO_2}, \tag{1}$$

where $[SO_2]_{aq}$ is the concentration of dissolved $SO_2$ in the aqueous phase, $p_{SO_2}$ the partial pressure of $SO_2$ in gas phase and $H_{SO_2}$ is Henry's law constant for $SO_2$, for which the expression and numerical parameters can be found in e.g., Sander (2015):

$$H_{SO_2}(T) = H_{SO_2}^0 \exp\left[B\left(\frac{1}{T} - \frac{1}{T^0}\right)\right], \text{with} \tag{2}$$
$$H_{SO_2}^0 = 1.3 \times 10^{-2}\,mol\,m^{-3}\,Pa^{-1}, \; B = 2900\,K \text{ and } T^0 = 298.15\,K \tag{3}$$

Aqueous $SO_2$ solution behaves like a weak acid, known as "sulfurous acid":

$$SO_{2(aq)} + H_2O \rightleftharpoons H^+ + HSO_3^-, \tag{R6}$$

with

$$\frac{[H^+][HSO_3^-]}{[SO_{2(aq)}]} = K_a^{H_2SO_3}, \tag{4}$$





with a weak acidity constant of $\mathrm{pK}_a^{\mathrm{H_2SO_3}} = 1.81$.

For the sake of completeness, it should also be mentioned that sulfurous acid can have a second acidic dissociation:

$$\mathrm{HSO_3^-} \rightleftharpoons \mathrm{H^+} + \mathrm{SO_3^{2-}} \tag{R7}$$

with $\mathrm{pK}_a^{\mathrm{HSO_3^-}} = 7.21$, but for pH values below 6 usually occurring in the atmosphere, this second dissociation hardly has an impact. In typical atmospheric conditions (including those found in volcanic plumes) with a pH between 2 and 7, aqueous S(+IV) is seen mainly in the form of $\mathrm{HSO_3^-}$ (Seinfeld and Pandis, 2006). One pathway for oxidation of S(+IV) to S(+VI) in aqueous phase is reaction of $\mathrm{HSO_3^-}$ with hydrogen peroxide $\mathrm{H_2O_2}$ (e.g. Shostak et al. (2019)):

$$\mathrm{HSO_{3(aq)}^-} + \mathrm{H_2O_{2(aq)}} \quad \rightarrow \quad \mathrm{SO_2OOH_{(aq)}^-} + \mathrm{H_2O} \tag{R8}$$

$$\mathrm{SO_2OOH_{(aq)}^-} + \quad \mathrm{H^+} \quad \rightarrow \quad \mathrm{H_2SO_{4(aq)}} \tag{R9}$$

However, in situations resembling volcanic plumes where $\mathrm{SO_2}$ is abundant, the availability of $\mathrm{H_2O_2}$ is a limiting factor for R8 and hence other reaction pathways become dominant (Pattantyus et al., 2018). In such cases, oxidation of $\mathrm{HSO_{3(aq)}^-}$ by $\mathrm{O_3}$ can become an important pathway (reaction R10; Lagrange et al., 1994; Seinfeld and Pandis, 2006; Pattantyus et al., 2018):

$$\mathrm{HSO_{3(aq)}^-} + \mathrm{O_{3(aq)}} \quad \rightarrow \quad \mathrm{H^+} + \mathrm{SO_{4(aq)}^{2-}} + \mathrm{O_{2(aq)}} \tag{R10}$$

Finally, oxidation of $\mathrm{HSO_{3(aq)}^-}$ by $\mathrm{O_2}$ with $\mathrm{Fe^{3+}}$ and $\mathrm{Mn^{2+}}$ as catalysts is another process that can be relevant in our case: (reaction R11; Connick and Zhang, 1996).

$$\mathrm{HSO_{3(aq)}^-} + \frac{1}{2}\mathrm{O_{2(aq)}} \quad \xrightarrow{Fe^{3+}, Mn^{2+}} \quad \mathrm{SO_{4(aq)}^{2-}} + \mathrm{H^+} \tag{R11}$$

The aim of this work is to estimate the sensitivity of $\mathrm{SO_2}$ conversion through these pathways in a volcanic plume to several parameters that remain poorly constrained.

In Section 2, we present the data used in the current study and the modelling choices that have been made. Section 3 presents the simulation outputs and their interpretation in terms of comparison to observations and in terms of sensitivity to multiple parameters. Finally, Section 4 draws conclusions and examines new perspectives that are not covered by the present study.

## 2   Material and methods

### 2.1   IASI instrument

The Infrared Atmospheric Sounding Interferometer (Clarisse et al., 2014, IASI) instrument onboard of Metop-A-C satellite series, the instrument is orbiting 817 km above the surface and provides a daily coverage of the earth with a pixel resolution of 12 km of diameter. IASI retrievals are widely used to observe and study the $\mathrm{SO_2}$ in the Earth's atmosphere Clarisse et al. (2012), including in volcanic plumes Carboni et al. (2012, 2016). This instrument has also been recently used to measure the Aerosol optical depth (AOD) of tropospheric volcanic sulphate particles (Guermazi et al., 2021).





## 2.2 CHIMERE model

The modelling work has been performed using v2020r1 version of the CHIMERE CTM (chemistry-transport model) (derived from v2020r1; Mailler et al., 2017; Menut et al., 2021) including new developments for vertical transport presented in Lachatre et al. (2020b); Mailler et al. (2020). The CHIMERE simulation domain covers the Central-Eastern mediterranean basin and

contains $874 \times 624$ cells at $2.25 \times 2.25 \, \mathrm{km}^2$ horizontal resolution. The geometry of the domain, which has a Lambert-conformal projection, was chosen to contain volcanic plume transport for a day, with a sufficiently fine resolution to resolve the volcanic plume during the first hours of its atmospheric advection (domain is displayed on Figure A1, in the appendix). At model resolution, the cell containing the vent has an average altitude of $2900 \, \mathrm{m.a.s.l.}$. The vertical distribution of the domain contains 40 layers, with the top of the domain being at $190 \, \mathrm{hPa}$. Horizontal advection in the CHIMERE model has been represented

using the classical Van Leer (1977) second-order slope-limited transport scheme.

Anthropogenic emissions are generated using the HTAP 2010 inventory (Janssens-Maenhout et al., 2015), boundary conditions for dust are calculated from GOCART global model (Ginoux et al., 2001) and from the global model LMDZ-INCA (Hauglustaine et al., 2004) for other species. The CHIMERE model has been forced using WRFv.3.7.1 (Weather Research and Forecasting Skamarock et al., 2008), with an update of the forcing meteorological variables every 20 minutes using the

WRF-CHIMERE online simulation framework (Briant et al., 2017; Menut et al., 2021). The WRF model has been run with 44 vertical levels starting from surface to 50 hPa with the same horizontal grid as the one used for CHIMERE. Large-scale meteorological fields used to force the WRF model at domain boundaries as well as for spectral nudging inside the simulation domain have been taken from the NCEP GFS dataset at $0.25°$ resolution (NCEP, 2015). The chemical modelling is as described in Mailler et al. (2017) (and references therein), including the reduced MELCHIOR2 chemical mechanism for inorganic chem-

istry, $SO_x$ chemistry , OH chemistry and more (Derognat et al., 2003; Menut et al., 2013). Gaseous oxidation pathways and aqueous oxidation through $O_3$ and $H_2O_2$ are included in this mechanism, and no modifications were made on this aspect of the chemistry mechanism. Oxidation of $SO_2$ by $O_{2(aq)}$ catalyzed by Fe and Mn is also available in the model; the evaluation of $[Fe^{3+}]$ and $[Mn^{2+}]$ have been adapted for the present study as discussed in Section 2.5.

## 2.3 Modelling Volcanic eruption emissions

The time and altitude profiles for the injection of $SO_2$ into the atmosphere (Table 1) were obtained using $SO_2$ emission flux rate measurement data from the ground-based DOAS FLAME (Differential Optical Absorption Spectroscopy FLux Automatic MEasurements) scanning network (e.g. Salerno et al., 2018). This method measures $SO_2$ fluxes during passive degassing, effusive and explosive eruptive activity using plume height inverted via an empirical relationship between plume height and wind speed (Salerno et al., 2009). In explosive paroxysmal events, such as in our case study, the plume is ejected to higher

altitudes and the linear height-wind relationship explained above cannot be utilized; therefore mass flux is retrieved in post-processing using the plume height estimated by visual camera and/or satellite observations.

On April 12, 2012, between 06 UTC and 16 UTC, a total $SO_2$ emission of $8.6 \, \mathrm{kt}$ was reported by this method. Emissions are localized around $8 \, \mathrm{km.a..s.l.}$. Volcanic $SO_2$ emissions are injected in the CTM with a skewed Gaussian profile (Eckhardt et al.,





2008; Mastin et al., 2009). The width of the Gaussian is defined by Full Width at Half Maximum (FWHM, equation 5) and it equals to 5 % of the center of injection's altitude ($\bar{x}$). The Gaussian profile is constructed with 13 altitude ranges, with widths corresponding to 1 % of the center of injection altitude.

$$\sigma = \frac{FWHM}{2.355} = \frac{\bar{x} \times 0.05}{2.355} \tag{5}$$

Volcanic emissions from explosive activities are more likely to be described with a skewed Gaussian profile (equation 6). In our case, we have selected a coefficient of skewness of $\alpha = 0.5\,\mathrm{m}$. Center of the injection is localized at $8\,000\,\mathrm{m.a.s.l.}$ ($\bar{x}$ value used to calculate $FWHM$); thus, $FWHM$ equals to $400\,\mathrm{m}$ and $\sigma^2$ to $170\,\mathrm{m}$ using equation 6. The vertical distribution is then constructed within 13 altitude ranges, with a width equal to $80\,\mathrm{m}$ (1 % of $\bar{x}$). The center of the injection is the center of the $7^{th}$ range. The entry for eruptive material in CHIMERE, before it is adapted to CHIMERE vertical grid, is displayed on Figure 1.

Water, Fe and Mn are emitted with the identical vertical distribution.

$$f(x) = \frac{\sqrt{2}}{\sqrt{\pi}(\sigma + \alpha)} \times \left[ e^{\frac{-(x-\bar{x})^2}{2\sigma^2}} \times \mathbf{1}_{]-\infty;\bar{x}]}(x) + e^{\frac{-(x-\bar{x})^2}{2\alpha^2}} \times \mathbf{1}_{]\bar{x};\infty[}(x) \right] \tag{6}$$

**Table 1.** $SO_2$ hourly flux ($\mathrm{kg.s^{-1}}$) estimates used as input for the CHIMERE model.

| date | time | $SO_2$ flux ($\mathrm{kg.s^{-1}}$) | $SO_2$ mass (t) | Fe mass (t) | Mn mass (t) |
|---|---|---|---|---|---|
| 12/04/2012 | 06 UTC | 249.9 | 899.6 | 0.78 | 0.077 |
| 12/04/2012 | 07 UTC | 400.9 | 1443.4 | 1.26 | 0.124 |
| 12/04/2012 | 08 UTC | 186.7 | 672.4 | 0.59 | 0.058 |
| 12/04/2012 | 09 UTC | 234.7 | 844.9 | 0.74 | 0.072 |
| 12/04/2012 | 10 UTC | 276.5 | 995.3 | 0.87 | 0.085 |
| 12/04/2012 | 11 UTC | 173.0 | 623.0 | 0.54 | 0.054 |
| 12/04/2012 | 12 UTC | 202.9 | 730.5 | 0.64 | 0.062 |
| 12/04/2012 | 13 UTC | 321.7 | 1158.2 | 1.01 | 0.099 |
| 12/04/2012 | 14 UTC | 199.1 | 716.9 | 0.63 | 0.061 |
| 12/04/2012 | 15 UTC | 144.4 | 519.9 | 0.45 | 0.044 |

## 2.4 Volcanic water emissions

Volcanic eruptions inject significant amounts of water in the atmosphere, particularly when considering the ambient humidity in the mid and upper troposphere. In the experiments containing volcanic water emissions, $H_2O$ is emitted similarly to $SO_2$

emissions, with identical time and vertical profiles as described in Section 2.3; To estimate the specific amount of the emitted

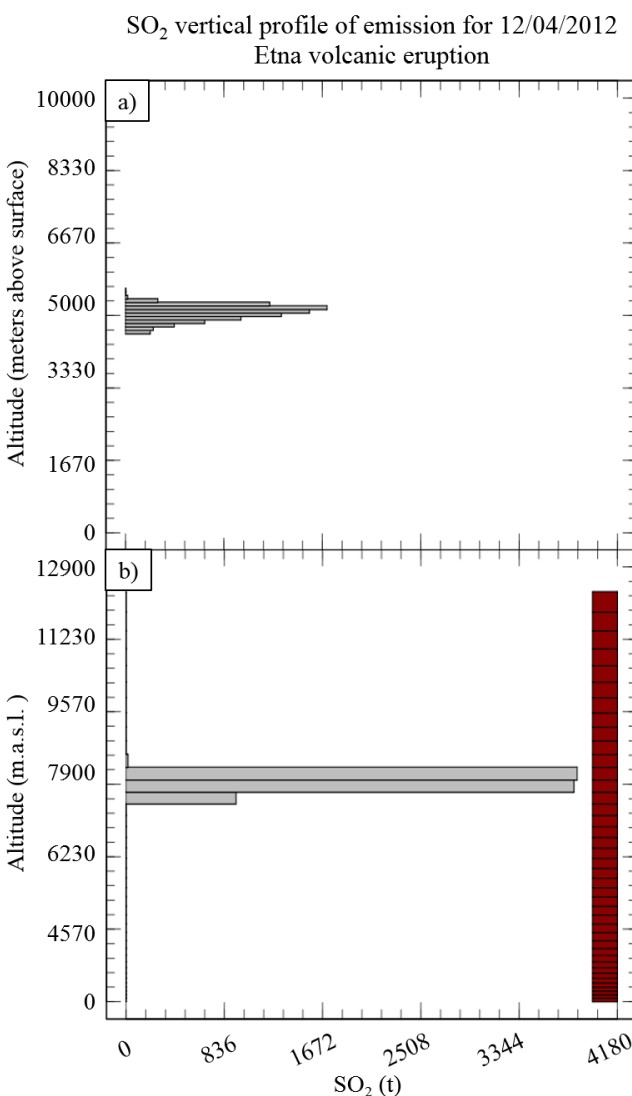

**Figure 1.** SO₂ vertical profile of emission for 12/04/2012 Etna volcanic eruption. a) Emission distribution following a skewed distribution. b) Emissions in the model after adaptation to the vertical grid used in CHIMERE (shown as red boxes).





$H_2O$, a molecular ratio between $SO_2$ and $H_2O$ has been implemented. For passive degassing, Shinohara et al. (2008) estimate the molecular $H_2O/SO_2$ ratio be around 45 when considering 27 events, and 26 when considering 13 events with the highest quality of data sampling, with a large variability. Nonetheless, explosive episodes eject water significantly larger amount, with $H_2O/SO_2$ molecular ratios likely reaching values of several hundreds, and the $H_2O/SO_2$ ratios associated to these events

still carry large uncertainties. To assess the sensitivity of our results to this ratio, we have tested several hypotheses for the $H_2O/SO_2$ ratio. As observational information is scarcely available for volcanic water emissions during explosive events, a central hypothesis of 300 molecules of $H_2O$ per molecules of $SO_2$ has been retained, corresponding to $725.6\,\mathrm{kt}_{H_2O}$.

Addition of volcanic water in the mid-troposphere can imply supersaturation. Consequently, in the model, water is added as water vapour until the partial pressure of water vapour reaches $105\,\%$ of the saturation vapour pressure $P_{H_2O}^{sat}$. Remaining

water emitted from the eruptive activity is added as liquid water or ice, depending on ambient temperature. Several studies have focused on the phase state of water in the upper troposphere (Textor et al., 2003; Hu et al., 2010; Kärcher and Seifert, 2016) and it is generally agreed that liquid water is virtually inexistent below the temperature of $\simeq 235\,\mathrm{K}$. Based on CALIOP measurements, Komurcu et al. (2014) has evaluated the supercooled liquid water fraction in clouds. Based on their Fig. 7, Eq. 7 gives a parabolic dependance of the supercooled liquid water fraction $H_2O_{(s)}$ on temperature, to be used between 235

and 273K (function is plot on Figure A2, in the appendix):

$$\%H_2O_{(s)} = -22.49 + 0.2092 \times T - 0.0004649 \times T^2 \tag{7}$$

## 2.5 Transition Metal Ions (TMI) dissolution into droplets and $[Fe^{3+}]_{(aq)}$ and $[Mn^{2+}]_{(aq)}$ threshold

In addition to $SO_2$ and water, volcanic eruptions inject significant amounts of transition metals into the atmosphere, such as Fe and Mn (Calabrese et al., 2011). These trace elements can be contained in the emitted volcanic ash, which is more relevant

during volcanic eruptive activity and must be considered in our simulations because of their catalytic role in the aqueous oxidation of $SO_2$ by $O_2$ (reaction R11). In our simulations, the amount of transition metals emitted by the volcano are defined relative to $SO_2$ emissions, with a molecular ratio of 1/1000 (i.e. 7.5 t) and 1/10000 (i.e. 0.73 t) respectively for Fe and Mn. The $SO_2$ oxidation to sulphate is catalyzed by $Mn^{2+}_{(aq)}$ and $Fe^{3+}_{(aq)}$ ions. The fraction of $Fe^{3+}$ in cloud droplets and available to the catalytic reaction is a complex matter which depends of several factors ($H_2S_{(g)}$ / $H_{2(g)}$ Hoshyaripour et al. 2014; Halogen

content $[Cl^-]$ Maters et al. 2017; ashes' surface and bulk compositions). In an ideal situation, up to a third of the total Fe on the ash surface can dissolve into the liquid phase coating volcanic particles, mostly in Fe(II) oxidation state (Galeazzo et al., 2018; Hoshyaripour et al., 2015). In our experiments, we consider 5 % of the total Iron and Manganese in the plume to be dissolved in the liquid phase (if clouds are produced) and therefore are available as catalysts for Reaction R11. In the case of liquid water in the plume, the Iron concentration in droplets ($mol.L^{-1}$) is calculated following the equation :

$$[Fe^{3+}_{(aq)}]_{i,j,l,t} = \frac{[Fe]_{i,j,l,t} \times 0.05 \times (1-icefrac_{i,j,l,t}) \times V_{i,j,l,t}}{M_{Fe} \times VH_2O_{i,j,l,t}} \tag{8}$$

where $i,j,l,t$ are the cell coordinates and time steps, $M_{Fe}$ the molar mass of Iron ($\mu g.mol^{-1}$), $VH_2O$ the volume of Super Cooled Liquid Water (SCLW; L), $V$ the volume of the cell ($m^3$), $[Fe_{(g)}]$ the concentration of Iron in the atmosphere ($\mu g.m^{-3}$)





and $ice frac$ is the fraction of ice cloud compared to liquid water (c.f. No Section 2.4).

The second parameter that needs to be fixed is the upper limit for $[Fe(III)]$. Seinfeld and Pandis (2006) indicate a range of Iron concentrations from 0.1 to $100\,\mu mol.L^{-1}$ in clouds, a large range compared to more recent studies, estimating $[Fe(III)]$ from 0.1 to $2\,\mu mol.L^{-1}$ (Maters et al., 2016, 2017). $Fe^{3+}$ is more likely to be dissolved in an acidic cloud droplet (pH below

2; Ayris and Delmelle, 2012), this particular condition can lead to $[Fe(III)]$ going up to $10\,\mu M$ (Ayris and Delmelle, 2012; Desboeufs et al., 2001). In our experiments, volcanic cloud droplets are particularly acidic, with a pH ranging from 1.5 to 3.5. In consequence, thresholds of $10\,\mu mol.L^{-1}$ for $[Fe(III)]$ and $1\,\mu mol.L^{-1}$ for $[Mn(II)]$ were chosen.

## 2.6    Simulations conducted and their purpose

Simulations have been organized into groups, to explore various parameters of interest. First, the simulations focused on the significance of gas phase conversion, aqueous phase conversion and transition metals as catalysts are gathered (Table 2). Then, we focused on the impact of volcanic water emitted during volcanic activity (Table 3). Next series of simulations focuses on evaluating initial parameters, such as the volcanic plume height of injection (Table 4). Finally, we have evaluated the plume chemistry sensitivity to transport modelling parameters, comparing two vertical advection schemes. These schemes

are described and tested in Lachatre et al. (2020b), however, the aforementioned article did not analyze their impact on the chemistry of the modelled plume.

Simulations underlined and labelled "Background" in Table 2,3,4,5 are simulations carried out without emissions originating from volcanic events; while they do not appear in figures themselves, these simulations are necessary to separate background information from our other sensitivity tests. In addition, to better understand the impact of ambient conditions alone on volcanic

$SO_2$ and $SO_{4(p)}^{2-}$ production, "Dry" simulations have been conducted in several cases. These simulations only include $SO_2$ as volcanic emissions; neither volcanic water nor metals are considered in these cases. For better readability of the results, a unique simulation labelled "Reference" is retained in every panel of simulations.

## 3    Results and Discussion

### 3.1    Reference simulation compared to IASI instrument

The background simulation has been used to exclude non-volcanic information from the Reference simulation and compare to the time-step surrounding IASI sounding for $SO_2$ (Figure 2). Although we have limited data, it is possible to extract several useful information from the comparison of IASI observations and CHIMERE simulations. Although the positioning of the plume on the simulation grid is not accurate in the simulations, other aspects of it, such as its thin shape and the concentration gradient seem to match the observations. These characteristics can be used to assess the quality of emissions data, while the

small discrepancies seen in the horizontal localisation of the plume can be linked to various parameters (meteorological fields, plume injection height, transport modelling scheme).




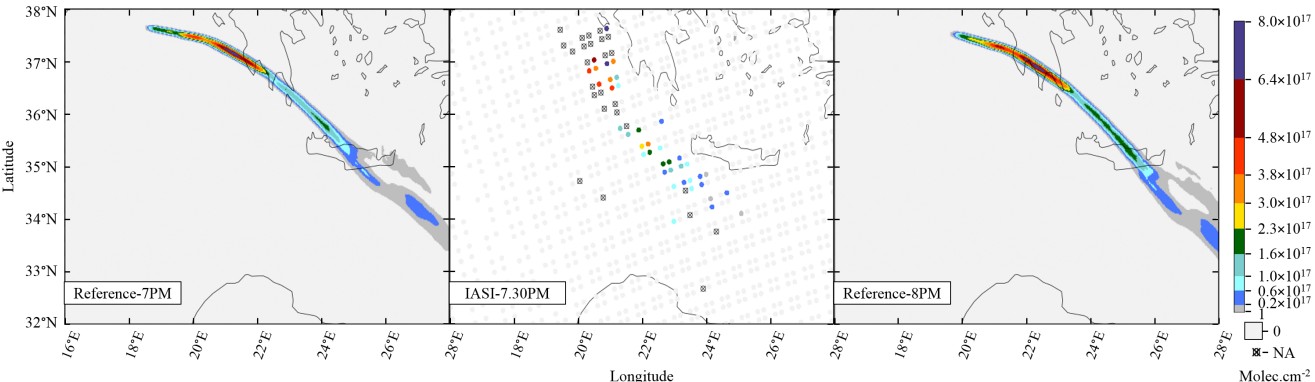

**Figure 2.** Columns of $SO_2$ from CHIMERE simulations and IASI measurements. IASI NAs data are shown as a crossed circle. Null values are shown in light gray. IASI sounding corresponds to 7.30PM UTC (center). CHIMERE Reference simulation is displayed for 7PM (left) and 8PM UTC (right).

### 3.2 Sensitivity tests for chemistry parameters

**Table 2.** Table gathering simulation parameters for sensitivity tests on chemistry parameters. Underlined items indicate background simulations that are used to retrieve background values.

| | | Sensitivity tests on chemistry parameters | | | | |
| --- | --- | --- | --- | --- | --- | --- |
| **Simulation label** | Volcanic $SO_2$ | Volcanic $H_2O$ | Volcanic TM | Volcanic clouds | Injection height | Vertical transport scheme |
| **Background** | 0.0 kt | 0.0 kt | 0.0 t | Not applicable | Not applicable | DL99 |
| **Dry** | 8.6 kt | 0.0 kt | 0.0 t | Not applicable | 8.0 km.a.s.l. | DL99 |
| **No SCLW** | 8.6 kt | 725.6 kt | 0.0 t | Not activated | 8.0 km.a.s.l. | DL99 |
| **No TM$_{aq}$** | 8.6 kt | 725.6 kt | 0.0 t | Activated | 8.0 km.a.s.l. | DL99 |
| **Reference** | 8.6 kt | 725.6 kt | 7.4 t | Activated | 8.0 km.a.s.l. | DL99 |

In the first group of tests (Table 2), the objective is to estimate the impact of various chemical pathways of $SO_2$ conversion. As mentioned in Section 2.6, Background and Dry simulations have also been conducted. For these simulations, since volcanic water is not emitted, super cooled water cannot be formed in the model; therefore, the table has been filled with "not applicable"

5 for relevant cases (e.g. volcanic clouds). The simulation No SCLW is slightly different. In this simulation, volcanic water is emitted, but SCLW is not formed from this water, therefore only the additional water vapour from the volcanic water is considered in the model chemistry and only gaseous pathway is evaluated. The next experiment, labelled as No TM$_{aq}$ (No Transition Metals in aqueous phase) is to evaluate the $SO_2$ conversion into liquid phase, without considering the pathway of oxidation by $O_2$ and catalyzed by $Fe$ and $Mn$; this is presented by Galeazzo et al. (2018) to be the main pathway of $SO_4^{2-}{}_{(p)}$ production.

10 The Reference simulation is considered to be the most realistic simulation performed in this work, in which $SO_2$ emissions are set to 8.6 kt, $H_2O$ emissions to 725.6 kt, transition metals emissions to 7.4 kt, volcanic super cooled liquid water clouds





are activated, mean injection height set to $8.0\,\mathrm{km}$ and the vertical transport scheme from Després and Lagoutière (1999) is used.

Figure 3 summarizes the results of simulations conducted in Table 2. Figure 3 shows the hourly evolution of a) the volcanic sulphates mass, b) of volcanic SCLW, d) the minimum volume containing $25\,\%$ of $SO_{4(p)}^{2-}$ mass, c) consequently, the AOD corresponding to the plume following the volume selection and e) the mass of hydroxyl radical. It can be seen that when volcanic water is added (simulations No $TM_{(aq)}$; Reference) SCLW is formed in the mid-troposphere which is a necessary element to evaluate aqueous chemistry paths. The Dry simulation allows us to evaluate the production of sulphates from reaction to background OH, and appears to be the main oxidation pathway in our experiment ($70\,\%$). The addition of volcanic water vapour without formation of SCLW did not significantly increase the conversion of $SO_2$ to $SO_{4(p)}^{2-}$. The same can be said about the addition of SCLW without TM (No $TM_{(aq)}$ simulation). However, the Reference simulation, which includes volcanic TM significantly increases the conversion of $SO_2$ ($25\,\%$). This additional formation of $SO_{4(p)}^{2-}$ is produced in a very small volume containing the volcanic cloud, which significantly change the optical properties of the plume (and eventually its radiative forcing generated) as it is shown by the evolution of plume's AOD. The comparison of the simulations conducted to understands the impact of the various chemical pathways has shown that the conversion of $SO_2$ mainly occurs in gas phase from reaction with the ambient OH ($70\,\%$) and then as a second pathway from the oxidation with $O_2$ catalyzed by TM in the aqueous phase ($25\,\%$).



**Figure 3.** Sensitivity tests on vertical transport scheme. a) $SO_{4(p)}^{2-}$ (kt), b) SCLW (kt), c) AOD for plume $\subset$ 25 % of $SO_{4(p)}^{2-}$ mass, d) Minimum volume (km$^3$) $\subset$ 25 % of $SO_{4(p)}^{2-}$ mass, e) OH radical (t).





**Table 3.** Table gathering simulation parameters for sensitivity tests on volcanic water emissions. Underlined items indicate background simulations that are used to retrieve background values.

| Sensitivity tests on volcanic water emissions | | | | | |
|---|---|---|---|---|---|
| **Simulation label** | Volcanic $SO_2$ | Volcanic $H_2O$ | Volcanic TM | Volcanic clouds | Injection height | Vertical transport scheme |
| **Background** | 0.0 kt | 0.0 kt | 0.0 t | Not applicable | Not applicable | DL99 |
| **Dry** | 8.6 kt | 0.0 kt | 0.0 t | Not applicable | 8.0 km.a.s.l. | DL99 |
| **WV200** | 8.6 kt | 483.7 kt | 7.4 t | Activated | 8.0 km.a.s.l. | DL99 |
| **Reference (WV300)** | 8.6 kt | 725.6 kt | 7.4 t | Activated | 8.0 km.a.s.l. | DL99 |
| **WV400** | 8.6 kt | 967.5 kt | 7.4 t | Activated | 8.0 km.a.s.l. | DL99 |

Then, our work focuses on the impact of various volcanic $SO_2/H_2O$ ratios (Table 3). Our Reference case takes a ratio of 1 molecule of $SO_2$ per 300 molecules of $H_2O$ emitted, thus 725.6 kt of $H_2O$ and 8.6 kt of $SO_2$. Two additional cases have been tested with volcanic $SO_2/H_2O$ ratios of 1/200 and 1/400, corresponding to 483.7 kt and 967.5 kt, and labelled WV200 and WV400 respectively. The aforementioned ratios are considered as threshold values for paroxysmal eruptions. The water

5   apportionment between its various physical states is displayed on Figure B1, in the appendix. Figure 4 summarizes the results of simulations conducted in Table 3. As the cloud generation is a threshold process, the amount of SCLW from volcanic eruption is not linearly linked to the volcanic water vapour emissions. In the WV300 scenario 1.3 $kt_{SCLW}$ is formed at peak time interval (around 12/04 at 12h), against 2.9 $kt_{SCLW}$ and 0.5 $kt_{SCLW}$ in the WV400 and WV200 scenarios respectively. The formation of sulphate with WV400 is 80 % stronger than in the simulation with no volcanic water, and 40% stronger than in

10   the simulation with WV200. This results in an increase of sulphates production; however, this is not a linear process either since the concentrations of $SO_2$ and the TM are different in the aqueous phase in each of the three cases. Thus, the amount of volcanic water impacts the optical properties of the plume, as the plume's AOD significantly increases following SCLW mass. This aspect is highlighted in the Figure 5 , which displays AOD spatial distribution and summarizes what is shown in Figure 4c-d for the 200 nm AOD.





**Figure 4.** Sensitivity tests on vertical transport schemes. a) $SO_{4(P)}^{2-}$ (kt), b) SCLW (kt), c) AOD for plume $\subset 25\%$ of $SO_{4(P)}^{2-}$ mass, d) Minimum volume ($km^3$) $\subset 25\%$ of $SO_{4(P)}^{2-}$ mass, e) OH radical (t).





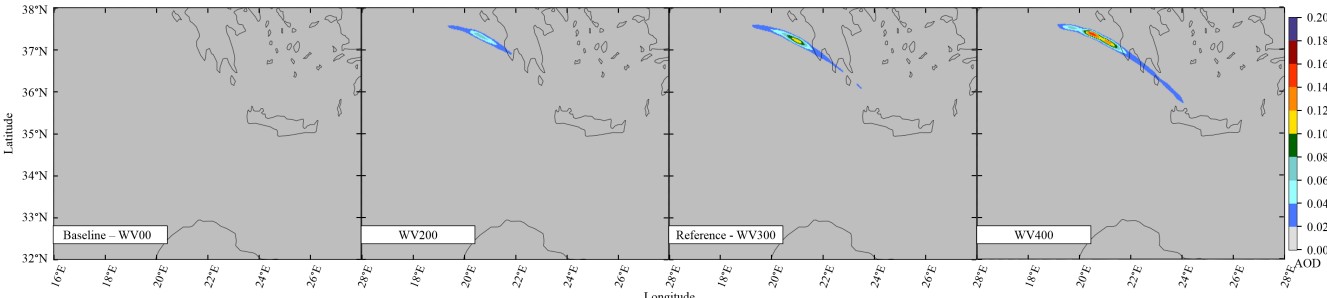

**Figure 5.** CHIMERE modelled 200 nm AOD from Volcanic sulphates.

**Table 4.** Table gathering simulation parameters for sensitivity tests on injection height. Underlined items indicate background simulations that are used to retrieve background values.

| Sensitivity tests on injection height | | | | | | |
|---|---|---|---|---|---|---|
| **Simulation label** | Volcanic $SO_2$ | Volcanic $H_2O$ | Volcanic TM | Volcanic clouds | Injection height | Vertical transport scheme |
| **Background** | 0.0 kt | 0.0 kt | 0.0 t | Not applicable | Not applicable | DL99 |
| **Dry** | 8.6 kt | 0.0 kt | 0.0 t | Not applicable | 8.0 km.a.s.l. | DL99 |
| **Reference** | 8.6 kt | 725.6 kt | 7.4 t | Activated | 8.0 km.a.s.l. | DL99 |
| **D.** **8.5 km** | 8.6 kt | 0.0 kt | 0.0 t | Not applicable | 8.5 km.a.s.l. | DL99 |
| **R.** **8.5 km** | 8.6 kt | 725.6 kt | 7.4 t | Activated | 8.5 km.a.s.l. | DL99 |

Next, to evaluate the impact of the surrounding environment (Table 4), we have conducted sensitivity tests on the plume's injection height. In our reference simulation, the plume injection is centered around 8.0 km.a.s.l.. A sensitivity test with the injection centered around 8.5 km.a.s.l. has been realized. This second test will provide an environment with a lower atmospheric pressure, lower temperature, dryer atmosphere and different wind speed and direction. Figure 6 summarizes the results of simulations described in Table 4. Comparing simulations Dry 8.0 km and Dry 8.5 km show differences in the results. In the Dry 8.5 km case, less SCLW is generated because of lower humidity, nonetheless more $SO_2$ are converted to sulphates. This is explained by the higher diffusion of the plume at higher altitude, as it can be seen that the minimum volume occupied by 25 % of the sulphate mass is bigger than in the Dry 8.0 km. Consequently, this higher dispersion allows a more efficient conversion from OH which is the limiting reactant in the gas phase oxidation.





**Figure 6.** Sensitivity tests on vertical transport scheme. a) $SO_{4(p)}^{2-}$ (kt), b) SCLW (kt), c) AOD for plume $\subset 25\%$ of $SO_{4(p)}^{2-}$ mass, d) Minimum volume (km$^3$) $\subset 25\%$ of $SO_{4(p)}^{2-}$ mass, e) OH radical (t).





**Table 5.** Table gathering simulation parameters for sensitivity tests on vertical transport scheme. Underlined items indicate background simulations that are used to retrieve background values.

| Simulation label | | Volcanic $SO_2$ | Volcanic $H_2O$ | Volcanic TM | Volcanic clouds | Injection height | Vertical transport scheme |
|---|---|---|---|---|---|---|---|
| **Background** | | 0.0 kt | 0.0 kt | 0.0 t | Not applicable | Not applicable | DL99 |
| **Dry** | | 8.6 kt | 0.0 kt | 0.0 t | Not applicable | 8.0 km.a.s.l. | DL99 |
| **Reference** | | 8.6 kt | 725.6 kt | 7.4 t | Activated | 8.0 km.a.s.l. | DL99 |
| **Background VL** | | 0.0 kt | 0.0 kt | 0.0 t | Not applicable | Not applicable | VL77 |
| **D.** | **VL** | 8.6 kt | 0.0 kt | 0.0 t | Not applicable | 8.0 km.a.s.l. | VL77 |
| **R.** | **VL** | 8.6 kt | 725.6 kt | 7.4 t | Activated | 8.0 km.a.s.l. | VL77 |

_Sensitivity tests on vertical transport scheme_

Finally, we investigate the impact of the plume dispersion on the computed chemistry (Table 5). Following the work done in Lachatre et al. (2020b) on different transport schemes, the Reference simulations has been conducted with the Després and Lagoutière (1999) vertical transport scheme (DL99), reducing the excessive plume diffusion that has been observed in previous work (Colette et al., 2011; Boichu et al., 2013; Lachatre et al., 2020a). In comparison, simulations with the Van Leer (1977)

(VL77) vertical transport scheme have been conducted; this transport scheme being expected to induce a larger spreading of the volcanic plume. For this last case, it was necessary to compute again a background simulation, using the VL77 vertical transport scheme.

Figure 7 summarizes the results of simulations conducted in Table 5. It was expected to see a larger spread of the plume in the sensitivity test with VL77 compared to the Reference simulation using the Després and Lagoutière (1999) vertical advection

scheme. Indeed, the volume of the plume has significantly increased, as it is displayed on Figure 7d). Consequently, less SCLW has been generated, but on the other hand, more sulphates were produced. This is due to higher conversion from the ambient OH, as it can be seen that more radical has been consumed (Figure 7e). It can also be noted that Reference VL's AOD is lower than Reference DL's AOD, due to the significantly larger spreading of Reference VL plume. This result was slightly unexpected as gaseous oxidation appeared to be linear at first; still, this new observation makes sense since more OH were

mobilized to react with the volcanic $SO_2$ in excess.



**Figure 7.** Sensitivity tests on injection height. a) $SO_{4(p)}^{2-}$ (kt), b) SCLW (kt), c) AOD for plume $\subset$ 25 % of $SO_{4(p)}^{2-}$ mass, d) Minimum volume (km$^3$) $\subset$ 25 % of $SO_{4(p)}^{2-}$ mass, e) OH radical (t).





## 4 Conclusions

In this study we aimed to investigate volcanic plume chemistry in the mid-troposphere region using the CHIMERE CTM. With the assistance of the IASI instrument's $SO_2$ sounding, we have determined that the CHIMERE model is able to reproduce a realistic structure for the plume as well as a correct intensity in terms of $SO_2$ columns after a number of assumptions were

made. Because of these encouraging preliminary observations, we gained confidence in the subsequent results. We have then analyzed the impact of various oxidation pathways by selectively shutting down these pathways to evaluate their contribution. For our study case, these sensitivity tests suggest that the main oxidation pathway is gas-phase oxidation by OH (about 70 %), followed by liquid-phase catalyzed oxidation by $O_2$ (about 25%). The fact that liquid-phase oxidation is dominated by TM-catalyzed oxidation is in line with the results of Galeazzo et al. (2018), and confirms that, unlike what typically happens in

polluted plumes, in such a volcanic plume availability of $H_2O_2$ rapidly becomes insufficient to substantially contribute to $SO_2$ oxidation. Therefore, our conclusion is that this oxidation pathway should be included in all modelling studies dealing with aqueous oxidation of volcanic $SO_2$. Our results calls for better constraining the quantities of dissolved $Fe^{3+}$ and $Mn^{2+}$ in volcanic cloud water through field measurements and further experimental studies.

We have tested the impact of $H_2O/SO_2$ ratio, with four hypotheses: No volcanic water, $H_2O/SO_2 = 200/1$, 300/1, and

400/1. These tests confirm that, depending on the $H_2O/SO_2$ ratio and on the background atmospheric condition, the presence and quantity of volcanic water vapour potentially has a strong impact on sulphate formation: in our case study, the formation of sulphate with $H_2O/SO_2 = 400/1$ is 80 % stronger than in the simulation with no volcanic water, and 40% stronger than in the simulation with $H_2O/SO_2 = 200/1$; therefore, in some cases such as mid-tropospheric plumes, including volcanic water may be necessary to correctly represent the conversion of volcanic $SO_2$ into sulphate aerosols. Apart from the above-mentioned

change in the overall quantity of sulphates formed, the localized formation of a liquid-containing volcanic plume may generate strong maxima in the sulphate AOD ($\sim$0.1 in our case study), while in the case devoid of volcanic water sulphate AOD never exceeds $\sim$0.005, far from any instrumental detection threshold. These sensitivity tests suggest the strong sensitivity of liquid-phase $SO_2$ oxidation to the injection height of the plume: if the plume is too low, then due to warm ambient temperature, volcanic water input may not be sufficient to reach saturation, but if the plume is too high, temperatures will be too cold to

permit the formation of a liquid aqueous phase. Therefore, our conclusion on the strong sensitivity of $SO_2$ oxidation to the $H_2O/SO_2$ ratio may hold only for mid-tropospheric plumes such as the one in our case study. This does not mean that the impact of volcanic water on chemistry is not relevant at higher altitudes: on the contrary, the influence of gas-phase volcanic water vapour may still be of interest in the case of upper tropospheric or stratospheric plumes, where an additional input of water vapour would enhance the formation of the OH.

Apart from the sensitivity to uncertainties concerning the physico-chemical processes and the forcings that we have discussed above, representation of $SO_2$ oxidation in volcanic plumes is also sensitive to the discretization strategies and to the numerical schemes that are used. For example, our sensitivity tests show that reducing excessive numerical diffusion by using an antidiffusive transport scheme such as Després and Lagoutière (1999) can change the structure of the modelled plume strongly, and in a complex way. In our case, reducing diffusion leads to a reduction in total production of sulphates, but with





sharper gradients and stronger peaks in concentration and AOD. Due to chemical nonlinearities (*e.g* the reduced availability of OH in the plume), reducing numerical diffusion can change the quantitative and qualitative properties of the resulting sulphate plume in a much more subtle way than just spreading the plume over a greater volume, as observed in Lachatre et al. (2020b) for an inert tracer. This confirms that chemistry-transport modellers should pay attention to reduce numerical diffusion in their

5    model, not only because excessive numerical diffusion will affect the spread of the plumes, but also, as we have shown here, because it will affect chemistry in a non-linear way, which in turn affects the AOD and therefore the radiative effect of particles.

This study shows the need to better constrain several parameters that we have shown to be crucial in the representation of the chemical behavior of volcanic plumes in the atmosphere. For example, it is critical to have better observational estimates of the $H_2O/SO_2$ ratio in an eruptive context. We also confirm the box-model results of Galeazzo et al. (2018), which suggest that the

10    impact of transition metals in liquid-phase oxidation of volcanic $SO_2$ is substantial. This highlights the need to better constrain volcanic emissions of Fe(III) and Mn(II) in the atmosphere and their subsequent repartition between volcanic ash and aqueous phase. The present study also highlights the need to find ways to reduce numerical diffusion in chemistry-transport models, through not only using better numerical strategies as shown here, but also by examining other approaches such as adaptative mesh refinement in both the horizontal and vertical dimensions.



**Code and data availability**

The source code for the CHIMERE model (Mailler et al., 2017) is available on: https://www.lmd.polytechnique.fr/chimere/. WRF source code is available on: https://github.com/wrf-model/WRF/. Clarisse, L. (2019). Daily IASI/Metop-B ULB-LATMOS sulphur dioxide (SO$_2$) L2 product (columns and altitude) [Data set]. AERIS. https://doi.org/10.25326/42. SO$_2$ (Salerno et al.,
2018) flux measurement data are available contacting the authors. Simulation outputs are available contacting the authors.

**Authors contributions**

All of the authors help to design the experiments. G.S. and S.G. provided the volcanic emissions data. A.C. prepared the anthropogenic emissions. S.M., G.S and M.L. adapted the model. M.L. carried out the simulations and prepared the results. M.L. and S.M. prepared the manuscript. All authors contributed to the text, interpretation of the results and reviewed the
manuscript.

*Acknowledgements.* This study has been supported by AID (Agence de l'Innovation de Défense) under grant TROMPET. Simulations have been performed on the Irene supercomputer in the framework of GENCI GEN10274 project. This work has been supported by the Programme National de Télédétection Spatiale (PNTS, http://www.insu.cnrs.fr/pnts ), grant n°PNTS-2019-9.

**Appendix A: Material and methods**

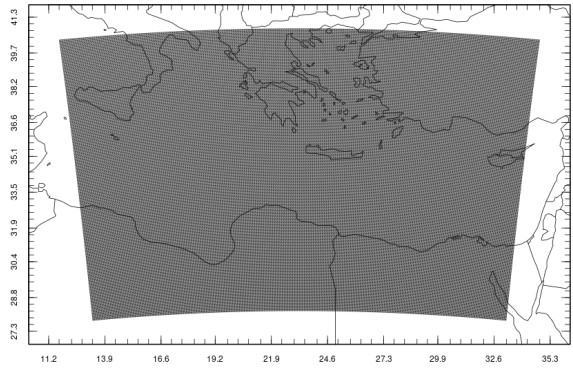

**Figure A1.** The CHIMERE simulation domain contains 874 × 624 cells at 2.250 km resolution

**Appendix B: Results and Discussion**



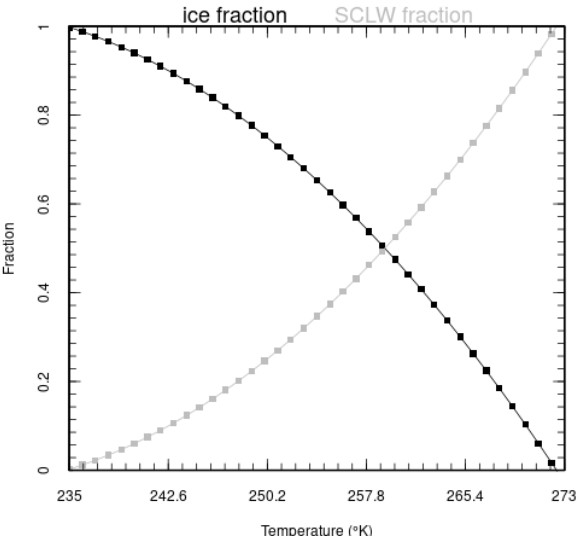

**Figure A2.** SCLW and ice fraction depending on temperature

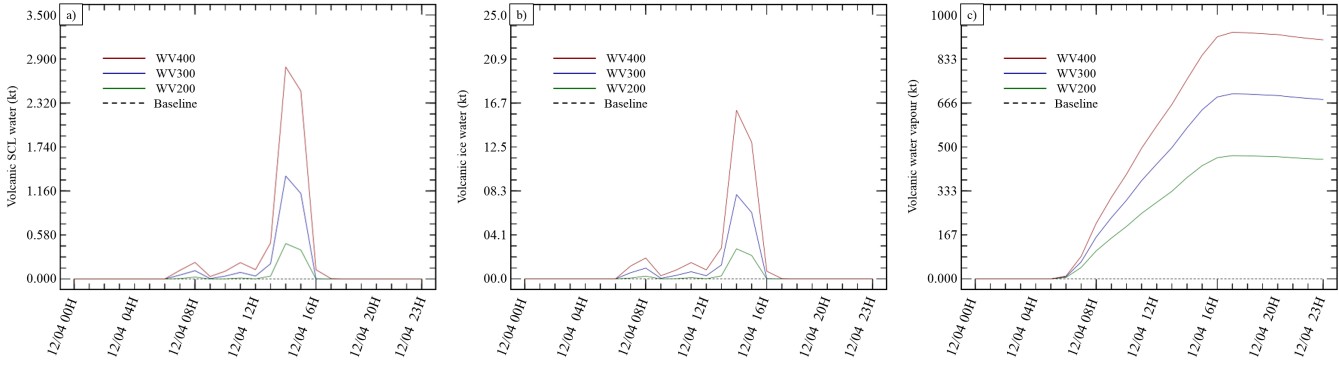

**Figure B1.** Water apportionment between its various states. a)SCL Water, b) ice, c) water vapour.



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
