# Peer review of "Modelling $SO_2$ conversion into sulphates in the mid-troposphere with a 3D chemistry-transport model: the case of Mount Etna's eruption on April 12, 2012."

_Atmospheric Chemistry and Physics, 2021_

## Author Comment (AC3)

**Modelling SO$_2$ conversion into sulphates in the mid-troposphere with a 3D chemistry-transport model: the case of Mount Etna's eruption on April 12, 2012:**
**Manuscrit submitted to *Atmos. Chem. Phys.**
**doi: 10.5194/acp-2021-1025-AC1**

**Answer to RCs**

Mathieu Lachatre, Sylvain Mailler, Laurent Menut, Arineh Cholakian,
Pasquale Sellitto, Guillaume Siour, Henda Guermazi, Giuseppe Salerno,
and Salvatore Giammanco

September 27, 2022

**1 Answers to RC #1**

We thank Anonymous Reviewer #1 for his comments showing that a more careful design for the figures and their captions was critical for clarity of the manuscript and Reader's understanding. We have answered below all the comments of the Reviewer, and hope our revised manuscript will meet the criteria for publication in *Atmos. Chem. Phys.*.

Below, text by the Reviewer appears *in italics*, our answers in **bold font**, and the description / retranscription of the changes brought to the manuscript appear **in blue**

**1.1 General comments**

*The manuscript presents a modeling investigation of SO2 conversion to sulfate in tropospheric volcanic plumes. As a case study, the authors simulate the explosive eruption of Mount Etna on Apr. 12, 2012, using the CHIMERE model and analyze the sensitivity of SO2 oxidation to water and transition metal contents, plume diffusion and plume altitude.*

*The results suggest that the dominant oxidation pathway is gas-phase oxidation by OH (about 70 %), followed by transition-metal catalyzed oxidation in aqueous phase (about 25%). The contribution of aqueous phase pathway increases with H2O/SO2 ratio and shows sensitivity to plume height as it modulates the liquid water content. The authors argue that H2O2 is rapidly diminished in volcanic plumes and cannot contribute to SO2 oxidation substantially.*

*This is a timely study with potential relevance for the atmospheric chemistry and volcanology communities. The manuscript is well structured and written. Nevertheless, there are some critical points:*

*[...]*

*Overall, I find the topic very important but am disappointed by the low quality of the presentation. Especially the negative OH mass shown in different figures challenges the scientific soundness of the study and the key outcome of the paper with respect to oxidation pathways. Therefore, I reject the manuscript. I hope this gives authors enough time to rigorously go through the methods, assumptions and experiments, critically validate their results and come up with a much better presentation quality.*

> **We have fixed the inconstitencies in Figure captions, and explained the** *negative OH mass* **issue, which was of course not negative in the model. This confusion was due to the fact that figure captions were unclear: negative numbers were actually the difference between one simulation with volcanic emissions and one simulation without.**

**1.2 "Critical points"**

*It seems that there is a significant flaw in the model as it allows negative mass for OH radicals. If the model allows this (which is fundamentally wrong), then it is no surprise that OH becomes the dominant pathway as it never diminishes. The authors are all well-known experts and should definitely check this in the experiments and results.*

*Also, in "other points":*

*Figure 3e: Negative mass?! This is wrong and challenges the whole study.*

> **As we have detailed in an earlier quick Author comment (doi: 10.5194/acp-2021-1025-AC1), there is no negative concentration of OH (or any chemical species) in the CHIMERE model. All the time series in the manuscript are differences from one simulation with volcanic eruption relative to the "Background simulation" without eruption.**
>
> **To clarify this matter, the following sentence has been added in the caption of all the figures containing time series :**
> **"All time series represent differences relative to the Background simulation (simulation without volcanic emissions)."**

*Methods lack some important information about the modelling setup, which complicates the reading and understanding of the results. Are you using online-coupled WRF-CHIMERE?*

> **From the submitted paper, p5 l16 : "the WRF-CHIMERE online simulation framework (Briant et al., 2017; Menut et al., 2021).", and the exact version number is also referred to (v2020r1). It is true that this version has several possibilities as regards interaction between chamistry and meteorology, allowing or forbidding retroaction of atmospheric composition on meteorology. To clarify the configuration we have used, the following sentence has been added :**

**For the present study, retroaction from atmospheric composition onto the WRF simulation is not activated, meaning that all the chemistry-transport simulations are forced by the exact same meteorological fields, which permits to isolate the physico-chemical effects of the volcanic eruption from the complex feedback it may have on the meteorological fields.**

*Besides, there is no information about AOD calculation.*
*When there are no observations to compare with or not radiative transfer calculations, why do we need AOD in the output? Why AOD at 200 nm? Are the aerosol microphysical processes in place? Or there is a simple assumption based on some particle size? Another principal question would be the role of AOD for model validation.*

**Aerosol Optical Depth (AOD) is a classic metric for the radiative impact of aerosols. It gives a qualitative and human-readable assessment of the importance of the aerosol column. The choice of $200\,\mathrm{nm}$ is arbitrary. With the relatively small diameter expected for freshly nucleated sulphate aerosol, the AOD is exprected to be stronger at such a short wavelength than in the visible wavelengths.**
**The following sentence has been added to the text:**

**For radiative processes, calculations are done online using the Fast-JX module version 7.0b (Bian and Prather [2002]). As described in Mailler et al. [2016], the CHIMERE model includes the feedback of aerosol layers on photolysis rates. The calculation of the Aerosol optical depths is also done by the Fast-JX module, using Mie calculations assuming spherical shape for all particles and external mixing.**

*I assume the plotted sulfate mass is calculated in the gas phase. But how does it come to aerosol phase?*

**The plotted sulphate mass is in the aerosol phase. At relatively high values of relative humidity, as it is the case of this plume (where RH is close to 100% in the core of the plume), binary nucleation of water + Sulphuric acid aerosol is a very efficient process so that the extremely hydrophilic sulphates tend to condensate quickly into a water-phase.**

**In all the figure captions, "$SO_{4(p)}^{2-}$ (kt)" has been replaced by "Sulphate aerosols ($SO_{4(p)}^{2-}$, kt)" in the figure captions to clarify this issue.**

*Figures need substantial improvement. For e.g. captions of figures 3 and 4 are wrong. Figure 2 is impossible to follow. Lines and colors need to be more visibal and distinguishable.*

**The figures have been improved for readability:**
**Figure 1 has been removed, and some figures were removed from the panels of former Figures 3,4,6,7 - now Figures 2,3,5,6.**
**Line width for all the time series have been increased for better readability, following referee coment.**
**Captions were also corrected in former Figure 3,4 and 6 - now Figures 2,3,5.**

*Design of experiments is fine, but the paper would be more beneficial if authors would present the chemistry parts only and leave out the sensitivity to the vertical transport scheme. The transport has to be validated first (based on available observations)*

The transport has been evaluated in dedicated paper of its own:

From the submitted paper,p5 l2: "including new developments for vertical transport presented in Lachatre et al. [2020]"

From the submitted paper,p9 l18: "These schemes are described and tested in Lachatre et al. [2020], however, the aforementioned article did not analyze their impact on the chemistry of the modelled plume."

From the submitted paper,p17 l8: "Following the work done in Lachatre et al. [2020] on different transport schemes"

Lachatre et al. [2020] has permitted to identify that using Després and Lagoutière [1999] for vertical advection gave better model performance. This has allowed us to choose this configuration as the reference configuration for the present study. Comparing this configuration with the more classical scheme of Van Leer [1977] permits to highlight the sensitivity of many processes on the representation of vertical advection, including for processes that could be thought linear such as gas-phase oxidation of $SO_2$ by $OH$. The following sentences have been added to the introduction to explain the possible relevance of performing these sensitivity tests:

From a modelling point of view, Lachatre et al. [2020] has shown that using the Després and Lagoutière [1999] antidiffusive advection scheme in the vertical direction rather than the classical order-2 Van Leer [1977] scheme substantially reduces plume diffusion, reducing plume volume and increasing its concentration. With this antidiffusive scheme, the plume volume is reduced by a factor ranging from 1.5 to 6 relative to the Van Leer [1977] scheme (depending on model configuration, see Lachatre et al. [2020] for details). Due to the many nonlinearities in the above-described physico-chemical mechanisms governing $SO_2$oxidation, the effect of such a change in numerical diffusion on the way the model represents sulphate formation is not straightforward: too much numerical diffusion in a model may enhance certain oxidation processes (such as oxidation by the background tropospheric species such as $OH$ or $H_2O_2$, which can be limited by the availability of these oxidants when the plume stays concentrated), and reduce others (such as aqueous-phase oxidation of $SO_2$) which can be favored by the simultaneous presence of large concentrations of volcanic water and volcanic $SO_2$. To examine these effects, it is also relevant to quantify the impact of these advection choices on the various oxidation paths of $SO_2$.

The conclusions drawn from these experiments were already present in the submitted version, which contribute to the justification of using these sensitivity experiments on transport schemes:

"Apart from the sensitivity to uncertainties concerning the physico-chemical processes and the forcings that we have discussed above, representation of $SO_2$ oxidation in volcanic plumes is also sensitive to the discretization strategies and to the numerical schemes that are used. For example, our sensitivity tests show that reducing excessive numerical diffusion by using an antidiffusive transport scheme such as Després and Lagoutière [1999] can change the structure of the modelled plume strongly, and in a complex way. In our case, reducing diffusion leads to a reduction in total production of sulphates, but with sharper gradients and stronger peaks in concentration and AOD. Due to chemical nonlinearities (*e.g* the reduced availability of $OH$ in the plume), reducing numerical diffusion can change the quantitative and qualitative properties of the resulting sulphate plume in a much more subtle way than just spreading the plume over a greater volume, as observed in Lachatre et al. [2020] for an inert tracer. This confirms that chemistry-transport modellers should pay attention to reduce numerical diffusion in their model, not only because excessive numerical diffusion will affect the spread of the plumes, but also, as we have shown here, because it will affect chemistry in a non-linear way, which in turn affects the AOD and therefore the radiative effect of particles."

*One critical aspect is the sensitivity to pH. The aqueous pathways strongly depend on pH. The authors should vary the pH in a range like 1-6 and then calculate the rate of sulfate formation for different pathways/experiments. I would like to see the plots for pH and $SO_2$.*

**The sensitivity to pH is indeed an important factor to aqueous chemistry. However, we would rather not directly set the pH value in the aqueous phase, as it is one of CHIMERE's abilites to calculate it in regard to chemical conditions. It also implies that the model does consider the impact of the pH in the aqueous phase chemistry. We agree that it could have been an additional test, but this was not the focus of our study.**

**1.3 Other points**

*P2L24: volcanic particle size distribution is evolving to a coarser distribution as time goes by not always. Volcanic PSD can move to finer modes if ash is involved. Upon ash removal from the atmosphere, the volcanic PSD moves from coarse to fine but increases later due to new particle formation.*

**the sentence as it appears in the Manuscript is indeed too general, and has the caveats highlighted by the Reviewer. The papers by Pianezze and Sahyoun on which this sentence is based, have actually a smaller focus, only focusing on $SO_2$ emissions without ash emissions, and the role of secondary sulphate aerosols formed in these plumes as CCN. The sentence in the text has been modified in a more careful way as follows:**

**Pianezze et al. [2019] and Sahyoun et al. [2019] have explored the role of secondary sulphate aerosols in the volcanic plumes from Etna and Stromboli, showing that these secondary aerosols are initially nucleated with very small diameters, but that their size distribution is evolving to a coarser distribution as time goes by so that these sulphate particles can serve as CCN far from the vent.**

*P2L35: It is not clear if the model takes into account aerosol-radiation interactions and its impacts on photolysis. This can substantially affect the OH generation (R1 and R2).*

**The aerosol-radiation impacts on photo-chemistry is considered in CHIMERE. This is now described with the sentence below:**

**For radiative processes, calculations are done online using the Fast-JX module version 7.0b (Bian and Prather [2002]). As described in Mailler et al. [2016], the CHIMERE model includes the feedback of aerosol layers on photolysis rates. The calculation of the Aerosol optical depths is also done by the Fast-JX module, using Mie calculations assuming spherical shape for all particles and external mixing.**

*P9L9: this title is odd. Use something like numerical experiments.*

**"Numerical experiments" is a correct vocabulary, as is "Numerical simulations". thefore we have changed the initial, possibly odd title "Simulations conducted and their purpose" to:**

**Description of numerical simulations**

*P9L25-30: Because of the low quality of Fig 2, it is not easy to follow the arguments here. Perhaps one option would be to superimpose IASI data on both plots (7 and 8 PM and use larger dots so one can better compare them).*

**We have not been able to improve this figure, which is at the best of our capabilities. satellite data is relatively scarce, and with many missing data in particular in critical areas around the edge of the plume. We think the difficulty in reading / interpreting this figure is not due to the figure itself, for which we did every effort to have a common area and color bars for model and satellite, but to the intrinsic limitation of data resolution and availability. Therefore, we have inserted a paragraph in the article to better explain and interoret this figure:**

Comparison of model outputs with satellite data from the IASI instrument (Figure 1) reveal that several aspects of the simulation outputs are consistent with observations. First, the general shape of the plume, with a NW-SE orientation, fits the observations. The range of values for $SO_2$ columns is also consistent as well as their structure, with weaker values in the southern part of the plume (around or below $2 \times 10^{17} \mathrm{molec\,cm^{-2}}$) and stronger values (above $5 \times 10^{17} \mathrm{molec\,cm^{-2}}$). Differences are also visible: the plume as represented by CHIMERE is shifted to the North and to the East compared to the plume as observed by IASI and the modelled plume extends further towards the southwest, which is not visible in the satellite data. Due to the lack of spatial continuity of the IASI data, it seems difficult to estimate a global mass of $SO_2$ in the plume. All in all, comparision with IASI data (Figure 1) confirms a correct localization and shape of the plume in CHIMERE (but with a horizontal offset of a few hundred kilometers), which is an indirect indication that the plume injection height in the model is correct: due to substantial wind shear in the troposphere, a large error in the injection height would result in a larger error in the position of the plume.

*P11L14: understand*

This was modified accordingly:

to understand the impact of the various...

*Figure3: The caption seems to be wrong. It should be sensitivity tests for chemistry, or? Lines are impossible to distinguish. Use thicker lines with colors that are better distinguishable.*

The thickness of all lines has been been increased for readability.

Former figures 3,4,5 captions were indeed incorrect, they have been modified and extended as follows for clarifications:

Figure 3 - Now 2: Sensitivity tests on chemistry parameters. a) Sulphate aerosols $(SO_{4(p)}^{2-}$, kt), b) Super Cooled Liquid Water (kt), c) Minimum volume $(km^3) \subset 25\%$ of $SO_{4(p)}^{2-}$ mass, d) AOD for plume $\subset 25\%$ of $SO_{4(p)}^{2-}$ mass. The Reference simulation is the closest to a realistic case. All time series represent differences relative to the Background simulation (simulation without volcanic emissions). Figure 4 - Now 3: Sensitivity tests on volcanic water emissions. a) Sulphate aerosols $(SO_{4(p)}^{2-}$, kt), b) Super Cooled Liquid Water (kt), c) Minimum volume $(km^3) \subset 25\%$ of $SO_{4(p)}^{2-}$ mass, d) AOD for plume $\subset 25\%$ of $SO_{4(p)}^{2-}$ mass. The Reference simulation is the closest to a realistic case. All time series represent differences relative to the Background simulation (simulation without volcanic emissions). Figure 6 - Now 5: Sensitivity tests on injection height. a) Sulphate aerosols $(SO_{4(p)}^{2-}$, kt), b) Super Cooled Liquid Water (kt), c) Minimum volume $(km^3) \subset 25\%$ of $SO_{4(p)}^{2-}$ mass, d) OH radical (t). The Reference simulation is the closest to a realistic case. All time series represent differences relative to the Background simulation (simulation without volcanic emissions).

*Figure 3d: What is the use of sulphate volume?*
The volume of the sulphate plume is used to calculate plume's AOD.
It is also helpful to understand $OH$ deplation in the "Sensitivity tests on injection height" and "Sensitivity tests on vertical scheme" experiments.

**2 Answers to RC #2**

We thank Anonymous Reviewer #2 for his careful reading of our article and useful suggestions, in particular to alleviate and improve the graphical content in terms of numbers of figures and their quality. We have submitted a revised version addressing the Reviewer's comments.

Below, text by the Reviewer appears *in italics*, our answers in **bold font**, and the description / retranscription of the changes brought to the manuscript appear in blue

**2.1 General comments**

*In this paper, Lachatre et al. attempt to test a model of $SO_2$ chemical conversion in a volcanic plume, focusing on a particular eruption of Mount Etna in Italy. The authors focus on one day, and follow the plume with the CHIMERE chemical transport model. Results show that sulfate is produced mainly through oxidation of $SO_2$ by OH in the gas phase (70%) and also by aqueous-phase oxidation of O2, catalyzed by Mn2+ and Fe3+ ions (25%). The authors test the model with different plume heights, water availability, and transport algorithms. Given the low abundance of $H_2O_2$ in volcanic plumes, better characterization of these $SO_2$ oxidation pathways is of interest. The work also has importance for better understanding the role of volcanic particles as cloud condensation nuclei and for better constraining radiative forcing of preindustrial climates. While not breathtakingly novel, the paper merits publication with minor revisions.*

**2.2 Minor issues**

*1. The reader would like to learn a bit more about why explosive volcanic eruptions emit large quantities of water. (Page 2, Line 30).*

**This a broad question in the field of volcanology, on the geological side, clearly out of the scope of the present study. Reference volcanology literature such as Turekian [2003], Fischer and Chiodini [2015] could give some general insights to this question.**

*2. More information on how plume heights were determined would be helpful. (Page 5, Lines 25-31).*

**As discussed and described in the cited study of Salerno et al. [2009], the method is in part empirical due to the insufficient coverage in scanner facilities: "Plume height could not be robustly determined for each scan (as described in more detail below) and therefore an empirical approach was adopted in which visual observations of the systematic variation in plume height was related to wind speed. This allowed us to use the wind speed to constrain the plume height. The initial estimates for the relationship between plume height and wind speed were refined empirically"**
**As already mentioned in the text (" an empirical relationship between plume height and wind speed"), this method is in part empirical, relying on expert eye and appreciation combined with satellite and *in situ* measurements. We are therefore not able to give more useful details in the study.**

*3. Figure 1. The figure is not interesting and should be move to the Supplement.*

**Figure 1 had the ambition to allow the reader to visualize the vertical discretization in the model and the way it affects the fine representation of a plume in the vertical direction.**

As suggested by the Reviewer, Figure 1 has been moved towards the Supplement and renamed as Fig. A1.

*4. The text states that useful information can be extracted from a comparison of model output and observations from IASI. (Page 9, Lines 26-27). Could a more quantitative comparison be made? True, the middle panel of Figure 2 looks something like the model results, but a more substantive comparison would be appreciated.*

Unfortunately, we have not been able to perform a more quantitative comparison between the satellite and model data. As visible on Fig. 2, satellite data are scarce, and many points in / close to the plume are detected as missing data (crosses in the figure). Therefore, we have reformulated the text with a more careful wording, and tried to explain in more detail what indications we obtain from this comparision, even though the scarcity of satellite data in this case does not permit much of a quantitative comparison. The modified text is as follows:

Comparison of model outputs with satellite data from the IASI instrument (Figure 1) reveal that several aspects of the simulation outputs are consistent with observations. First, the general shape of the plume, with a NW-SE orientation, fits the observations. The range of values for $SO_2$ columns is also consistent as well as their structure, with weaker values in the southern part of the plume (around or below $2 \times 10^{17} \mathrm{molec\,cm^{-2}}$) and stronger values (above $5 \times 10^{17} \mathrm{molec\,cm^{-2}}$). Differences are also visible: the plume as represented by CHIMERE is shifted to the North and to the East compared to the plume as observed by IASI and the modelled plume extends further towards the southwest, which is not visible in the satellite data. Due to the lack of spatial continuity of the IASI data, it seems difficult to estimate a global mass of $SO_2$ in the plume. All in all, comparision with IASI data (Figure 1) confirms a correct localization and shape of the plume in CHIMERE (but with a horizontal offset of a few hundred kilometers), which is an indirect indication that the plume injection height in the model is correct: due to substantial wind shear in the troposphere, a large error in the injection height would result in a larger error in the position of the plume.

*5. Table 2 and all tables. Please identify the acronyms in a footnote. Also state in the caption of the Tables which simulation is the most realistic.*

We thank the Reviewer for drawing our attention to the numerous and maybe confusing acronyms. The acronyms are now explained in each table caption, and summarized here in Tab. 1. It has been stated in each table caption which simulation was the closest to a realistic case.

"For better readability of the results, a unique simulation labelled "Reference" is retained in every panel of simulations. A table describing all the simulation was added in appendix in Table A1 along with the Figure A5 who sums up experiments' results." Captions of tables 2, 3, 4, 5 have been rewritten to explain all the acronyms present in the tables. Table 1 has been added

*6. Figure 3 and most of the figures. There is an abundance of figures in the paper that do not seem to contribute much to the overall message. I recommend that the authors choose one panel from each Figure like Figure 3 and keep these panels in the main text. (Likley these will be the panels showing the timeseries of volcanic sulfate mass.) Put the remaining panels, if deemed necessary, in the Supplement. Also, the thin lines could be fattened to improve the visual message, and less white should be retained in the panels. Having so many figures with so much white space dilutes the main messages of the paper. The figures in the appendix can also go in the Supplement.*

We have tried to reduce tha amount of panels and the space they occupy, as suggested by the Reviewer. We have selected the most informative figures for each panels, which are not more homogenous but this has allowed in a gain of space and readability.

For figures 3, 4, 6, and 7 of the initial manuscript (2, 3, 5 and 6 of the revised manuscript), the number of subpanels has been brought down from 5 to 4, conserving only for each panel the total sulphate mass (tu illustrate sulphate formation). This allow a gain of space of $33\,\%$ compared to previous disposition.

*7. Captions for Figure 3 and similar figures. Please write out, without symbols or acronyms, what is being presented. Many readers will just browse through the paper and look most closely at the captions.*

Table 1: Synthetic list of the 13 simulations that have been performed for this study and their description

| Simulation label | Developed description |
| --- | --- |
| **Background** | Simulates the atmosphere as it would be without a volcanic eruption (no emissions of volcanic $SO_2$, volcanic water or volcanic transition metals |
| **Dry** | Simulates the atmosphere as it would be with emissions of volcanic $SO_2$ but no emissions of volcanic water or volcanic transition metals |
| **No SCLW** | Simulates the atmosphere as it would be with emissions of volcanic $SO_2$ and of volcanic water but volcanic water is not allowed to contribute to a liquid phase. No emissions of transition metals.
 "No SCLW" stands for "No Super Cooled Liquid Water" |
| **No TM$_{aq}$** | Simulates the atmosphere as it would be with emissions of volcanic $SO_2$ and of volcanic water. Volcanic water is allowed to contribute to a liquid phase. No emissions of transition metals.
 "No TM$_{aq}$" stands for "No Transition Metals in Aqueous phase" |
| **Reference** | **Reference simulation**, inclusing emissions of volcanic $SO_2$, volcanic water and volcanic transition metals, and permitting volcanic water to contribute to a liquid phase. The $\frac{H_2O}{SO_2}$ mass emission ratio is set to 300. |
| **WV200** | Same as **Reference** but the $\frac{H_2O}{SO_2}$ mass emission ratio is set to 200.
 "WV200" stands for "Water Vapor 200" |
| **WV400** | Same as **Reference** but the $\frac{H_2O}{SO_2}$ mass emission ratio is set to 400.
 "WV400" stands for "Water Vapor 400" |
| **Dry 8.5 km** | Same as **Dry** but the volcanic plume is released at 8500 m.a.s.l instead of 8000 m.a.s.l |
| **Reference 8.5 km** | Same as **Reference** but the volcanic plume is released at 8500 m.a.s.l instead of 8000 m.a.s.l |
| **Background VL** | Same as **Background** but using the Van Leer [1977] advection scheme instead of Després and Lagoutière [1999] |
| **Dry VL** | Same as **Dry** but using the Van Leer [1977] advection scheme instead of Després and Lagoutière [1999] |
| **Reference VL** | Same as **Reference** but using the Van Leer [1977] advection scheme instead of Després and Lagoutière [1999] |

We have added a memo regarding the Reference simulation as the closest to a realistic case. We have explicited SCLW acronym.

**a) $SO_{4(p)}^{2-}$(kt), b) Super Cooled Liquid Water (kt), c) Minimum volume ($km^3$) $\subset$ 25 % of $SO_{4(p)}^{2-}$ mass, d) $OH$ radical (t). The Reference simulation is the closest to a realistic case.**

*8. The Conclusions section mentions the need to better constrain emissions of $Fe^{3+}$ and $Mn2+$ from volcanoes. (Page 20, lines 10-13). Do such emissions vary greatly among different volcanoes? What steps can be taken to constrain these emissions?*

There is not much data on $Fe^{3+}$ and $Mn^{2+}$ emissions from volcanoes: this is why we need them and we hope we will get them in future studies specifically focused on this topic. Based on what is known from the thermodynamic equilibria in high-Temperature gas-ash interaction Hoshyaripour et al. [2014], emissions of these metals can vary greatly among different volcanic environments (even by orders of magnitude) and hence their quantitative assessment is needed in order to better constrain our simulations. Next steps in better constraining Fe and Mn emissions at Mt. Etna are, in first place, to carry out more sampling and analysis of their contents in ash samples, and then in plume samples, through use of passive traps (filter-packs, moss-bags, etc.). The latter approach has to be performed in a systematic way, in order to capture plume emissions all around the summit craters and at different distances from them, both during passive degassing periods and during eruptions.
The following sentence has been added into the conclusion:

**As it has been shown by Hoshyaripour et al. [2014], emissions of these transition metals can vary greatly among different volcanic environments. Therefore, our results calls for better constraining the quantities of dissolved Fe(III) and Mn(II) in volcanic cloud water through field measurements and further experimental studies. In the case of Mount Etna, this quantification could be performed with more sampling and analyses of their contents in ash and plume samples through use of passive traps. To be comprehensive, this sampling would have to be done ad different distances from the craters, both during passive degassing periods and during eruptions.**

**References**

Huisheng Bian and Michael J. Prather. Fast-j2: Accurate simulation of stratospheric photolysis in global chemical models. *Journal of Atmospheric Chemistry*, 41(3):281–296, 2002. doi: 10.1023/A:1014980619462. URL https://doi.org/10.1023/A:1014980619462. 3, 5

Bruno Després and Frdric Lagoutière. Un schma non linéaire anti-dissipatif pour l'équation d'advection linéaire. *Comptes Rendus de l'Académie des Sciences - Series I - Mathematics*, 328(10):939 – 943, 1999. ISSN 0764-4442. doi: https://doi.org/10.1016/S0764-4442(99)80301-2. URL http://www.sciencedirect.com/science/article/pii/S0764444299803012. 4, 9

Tobias P. Fischer and Giovanni Chiodini. Chapter 45 - volcanic, magmatic and hydrothermal gases. In Haraldur Sigurdsson, editor, *The Encyclopedia of Volcanoes (Second Edition)*, pages 779–797. Academic Press, Amsterdam, second edition edition, 2015. ISBN 978-0-12-385938-9. doi: https://doi.org/10.1016/B978-0-12-385938-9.00045-6. URL https://www.sciencedirect.com/science/article/pii/B9780123859389000456. 7

G. Hoshyaripour, M. Hort, B. Langmann, and P. Delmelle. Volcanic controls on ash iron solubility: New insights from high-temperature gasash interaction modeling. *Journal of Volcanology and Geothermal Research*, 286:67 – 77, 2014. ISSN 0377-0273. doi: https://doi.org/10.1016/j.jvolgeores.2014.09.005. URL http://www.sciencedirect.com/science/article/pii/S0377027314002844. 10

M. Lachatre, S. Mailler, L. Menut, S. Turquety, P. Sellitto, H. Guermazi, G. Salerno, T. Caltabiano, and E. Carboni. New strategies for vertical transport in chemistry transport models: application to the case

of the mount etna eruption on 18 march 2012 with chimere v2017r4. *Geoscientific Model Development*, 13 (11):5707–5723, 2020. doi: 10.5194/gmd-13-5707-2020. URL `https://gmd.copernicus.org/articles/13/5707/2020/`. 4

S. Mailler, L. Menut, A. G. di Sarra, S. Becagli, T. Di Iorio, B. Bessagnet, R. Briant, P. Formenti, J.-F. Doussin, J. L. Gómez-Amo, M. Mallet, G. Rea, G. Siour, D. M. Sferlazzo, R. Traversi, R. Udisti, and S. Turquety. On the radiative impact of aerosols on photolysis rates: comparison of simulations and observations in the lampedusa island during the charmex/adrimed campaign. *Atmospheric Chemistry and Physics*, 16(3):1219–1244, 2016. doi: 10.5194/acp-16-1219-2016. URL `https://acp.copernicus.org/articles/16/1219/2016/`. 3, 5

J. Pianezze, P. Tulet, B. Foucart, M. Leriche, M. Liuzzo, G. Salerno, A. Colomb, E. Freney, and K. Sellegri. Volcanic plume aging during passive degassing and low eruptive events of etna and stromboli volcanoes. *Journal of Geophysical Research: Atmospheres*, 124(n/a), 2019. doi: 10.1029/2019JD031122. URL `https://agupubs.onlinelibrary.wiley.com/doi/abs/10.1029/2019JD031122`. 5

Maher Sahyoun, Evelyn Freney, Joel Brito, Jonathan Duplissy, Mathieu Gouhier, Aurlie Colomb, Regis Dupuy, Thierry Bourianne, John B. Nowak, Chao Yan, Tuukka Petj, Markku Kulmala, Alfons Schwarzenboeck, Cline Planche, and Karine Sellegri. Evidence of new particle formation within etna and stromboli volcanic plumes and its parameterization from airborne in situ measurements. *Journal of Geophysical Research: Atmospheres*, 124(10):5650–5668, 2019. doi: https://doi.org/10.1029/2018JD028882. URL `https://agupubs.onlinelibrary.wiley.com/doi/abs/10.1029/2018JD028882`. 5

G.G. Salerno, M.R. Burton, C. Oppenheimer, T. Caltabiano, D. Randazzo, N. Bruno, and V. Longo. Three-years of so2 flux measurements of mt. etna using an automated uv scanner array: Comparison with conventional traverses and uncertainties in flux retrieval. *Journal of Volcanology and Geothermal Research*, 183(1):76 – 83, 2009. ISSN 0377-0273. doi: https://doi.org/10.1016/j.jvolgeores.2009.02.013. URL `http://www.sciencedirect.com/science/article/pii/S0377027309000791`. 7

K.K. Turekian. Treatise on geochemistry. In H.D. Holland, editor, *Treatise on Geochemistry*. Elsevier, 2003. ISBN 978-0-08-043751-4. URL `https://www.sciencedirect.com/referencework/9780080437514/treatise-on-geochemistry`. 7

B. Van Leer. Towards the ultimate conservative difference scheme. iv. a new approach to numerical convection. *Journal of Computational Physics*, 23(3):276 – 299, 1977. ISSN 0021-9991. doi: https://doi.org/10.1016/0021-9991(77)90095-X. URL `http://www.sciencedirect.com/science/article/pii/002199917790095X`. 4, 9